# Natural Patterns in the Dawn and Dusk Choruses of a Neotropical Songbird in Relation to an Urban Sound Environment

**DOI:** 10.3390/ani14040646

**Published:** 2024-02-17

**Authors:** Noelia Bustamante, Álvaro Garitano-Zavala

**Affiliations:** 1Carrera de Biología, Universidad Mayor de San Andrés, La Paz P. O. Box 10077, Bolivia; 2Instituto de Ecología, Universidad Mayor de San Andrés, La Paz P. O. Box 10077, Bolivia

**Keywords:** acoustic communication, anthropogenic noise, behavior, bird song, chorusing, urban ecology, *Turdus*

## Abstract

**Simple Summary:**

Urban noise imposes significant challenges for the acoustic communication of birds that are able to survive inside cities. Birds sing most intensely during two periods of the day called dawn and dusk choruses, and although various responses to the urbanization of dawn songs have been reported, very little attention was paid to the dusk chorus. Our objective is to evaluate in urban and non-urban populations of the Chiguanco Thrush (*Turdus chiguanco*), a very common bird in the city of La Paz (Bolivia) if there are differences between the songs of both choruses, and if these variations are altered by urbanization. Our results show that the loudness, frequency range and number of songs per individual are greater in the dawn chorus and that urban individuals must increase the frequency and loudness of their songs in both choruses to cope with urban noise. Urban Chiguanco Thrushes even produce less than half as many songs per individual compared to non-urban individuals, probably due to the high cost of increasing loudness and frequency. If wild birds are forced to modify their songs so much within the city, this should alert us to the possible negative effects of urban noise on human health.

**Abstract:**

Urbanization is one of the more important phenomena affecting biodiversity in the Anthropocene. Some organisms can cope with urban challenges, and changes in birds’ acoustic communication have been widely studied. Although changes in the timing of the daily organization of acoustic communication have been previously reported, there is a significant gap regarding possible variations in song structure between dawn and dusk choruses. Considering that urbanization imposes different soundscapes for dawn and dusk choruses, we postulate two hypotheses: (i) there are variations in song parameters between dawn and dusk choruses, and (ii) such parameters within the city will vary in response to urban noise. We studied urban and extra-urban populations of Chiguanco Thrush in La Paz, Bolivia, measuring in dawn and dusk choruses: song length; song sound pressure level; minimum, maximum, range and dominant frequency; and the number of songs per individual. The results support our two hypotheses: there were more songs, and songs were louder and had larger band widths at dawn than at dusk in urban and extra-urban populations. Urban Chiguanco Thrushes sing less, the frequency of the entire song rises, and the amplitude increases as compared with extra-urban Chiguanco Thrushes. Understanding variations between dawn and dusk choruses could allow for a better interpretation of how some bird species cope with urban challenges.

## 1. Introduction

Urban growth is one of the most intense and severe environmental alterations of the Anthropocene that imposes challenges from individual species to eco-evolutionary processes [1]. As a result, biodiversity richness is reduced and species composition is altered within cities compared to non-urbanized landscapes [2,3,4,5].

These changes in biodiversity are partly explained by the differential potential of individual species to respond to urban disturbances [6,7,8,9,10]. Behavioral plasticity allows animal species to rapidly change their phenotype in response to environmental modifications imposed by urbanization [11,12,13], and acoustic communication in birds is one of the most studied behavioral changes related to response to urbanization [8,14,15,16,17].

Urban birds could change the structural and temporal parameters of their songs to cope with the challenges posed by urban noise. Among the reported structural changes is the increase in song frequency (e.g., [14,18,19,20]; but see [21,22]), the increase in the sound pressure level (loudness) of the song [17,23,24], and changes in the repertoire [15,25]; but see [22]. The most frequently reported temporal changes are those related to schedule adjustment that allows birds to avoid the noisiest periods [26,27]; but see [28,29,30] in relation to the effect of light pollution timing on bird songs.

In the daily organization of song production, several bird species display two special periods called dawn and dusk choruses, in which multiple individuals of the same or different species sing at the same time with greater intensity compared to other periods [27,31]. This phenomenon, at least for the dawn chorus, was described for temperate-zone and tropical bird species [31]. Such daily patterns imply special functions of songs shaped by natural selection, and their ecological and adaptive function is the subject of current discussion [31,32,33]. Although it is expected that the soundscape may vary naturally between dawn and dusk and that the sound parameters of the songs could change accordingly, to our knowledge, no studies have been conducted to compare such potential changes between dawn and dusk choruses. As “natural”, we refer to those adapted before the settlement of cities. Furthermore, cities do not necessarily follow the same temporal variations in the soundscape as in nature, and although several studies have been conducted to understand the effects of urbanization on dawn chorus [27], the effects of urbanization on the natural variations between dawn and dusk choruses have not been evaluated before.

Our objective is to evaluate such possible variations in a bird species that was capable of colonizing cities. We select the Chiguanco Thrush (*Turdus chiguanco*), an Andean bird species reported as one of the most ubiquitous, frequent, and abundant inhabitants, from the highly urbanized areas to the rural areas of La Paz, a high-altitude tropical city in South America [34]. For this species, in addition, behavioral responses to urbanization were previously reported [35]. This species performs striking choruses at dawn and dusk during the bird-breeding season in the city of La Paz and its surroundings. Furthermore, the study of species of the genus *Turdus* also allows comparison of responses to urbanization with congeneric species (e.g., [19,25,36,37,38,39,40]).

We compared several acoustic parameters of the songs performed in choruses at dawn and dusk by adult individuals that inhabit the highly urbanized areas of the city with those of individuals from extra-urban habitats. The parameters we measured were song length; song sound pressure level; minimum, maximum, range and dominant frequency; and the number of songs per individual. Considering that under natural conditions, dawn and dusk choruses occur in different soundscapes and urbanization imposes different soundscapes for bird communication, we postulate two hypotheses. (1) There are variations in song parameters between dawn and dusk choruses, and (2) The sound parameters of the songs within the city will vary compared to extra-urban songs for both the dawn and dusk choruses in relation to urban noise, measured as environmental sound pressure level.

## 2. Materials and Methods

### 2.1. Study Area

The city of La Paz is located between 16°26′ to 16°33′ S, 68°02′ to 68°10′ W, and 3300 to 4100 m in the Andes of Bolivia (Figure 1), in 2012 it occupied an area of 70 km^2^, and had no more than a million inhabitants [41]. As a sprawling city it grows at the expense of agricultural fields and natural vegetation on the hillsides. Natural vegetation is severely altered within the city, and the effect of urbanization on native bird richness is comparatively worse relative to other South American cities [42]. For the urban gradient we used two of the four categories proposed by Marzluff et al. (2001; [43]), “urban” and “rural/exurban”. But instead using “rural” or “exurban”, we have used the term “extra-urban” because it includes all non-urban systems [44] and is best applied to the city of La Paz, where outside the city there is a mixture of natural and agricultural matrices. From now on we will call both categories as “zones”. The urban zone had more than 50% impervious surfaces, with small and scarce public and private green areas dominated by exotic plant species, high rates of pedestrian and vehicular traffic, multi-family houses and multi-story buildings with a density of approximately 120 inhabitants per km^2^. The extra-urban zone had less than 20% buildings, dominated by a matrix of secondary shrub vegetation of native species mixed with small agricultural fields and exotic trees.

### 2.2. Spatial and Temporal Design

We worked during the bird-breeding period that occurs in La Paz during the rainy season, between December 2011 to March 2012. We selected one-hectare plots in urban and extra-urban zones with relatively similar abundance of Chiguanco Thrush (between 15 to 20 individuals per hectare) in order to control for the potential effect of density on song parameters [45,46]. Although it is most likely that males produced all the songs because there is no evidence that females sing in the genus [47,48], it is possible that females also sang as there increasing evidence for this [49] and because both sexes in this species are very similar. We determined abundance before selecting plots; to do this we established a 10 min counting point in several potential plots, evaluating each plot one day in the morning (08:00 to 08:30). From this, we selected three plots per zone, the urban ones were two squares (U1 and U2), and a main avenue with its adjacent streets (U3) (Figure 1), and the extra-urban plots were a municipal protected area (E1), a natural area adjacent to recent human settlements (E2), and the campus of the Universidad Mayor de San Andrés (E3). The latter could be considered a peri-urban plot but its soundscape convinced us to consider it suitable for our purposes. Between urban and extra-urban zones there were differences in the soundscape with a minimum average of 65 dB for urban plots and a maximum average of 60 dB for extra-urban plots [50]. Specific information for each study plot appears in Appendix A.

Before starting the measurements, in December 2011, we determined the daily song production schedule of the Chiguanco Thrush by visiting the selected plots from 04:30 in the morning (before the sunrise) to 19:30 in the afternoon (after the sunset). We determined that the best time to record the dawn chorus was from 05:00 to 07:00 and for the dusk chorus from 17:00 to 19:00. We obtained data from January to March 2012 visiting each plot twice, avoiding rainy or windy days, one day in the morning for the dawn chorus and a different day in the afternoon for the dusk chorus. We randomized the order of visits to each plot and period of the day.

### 2.3. Obtaining Song Sound Parameters and Environmental Sound Pressure Level

We recorded the majority of individuals singing in each plot per day, for one hour, along with their respective song sound pressure level (SSP) and environmental sound pressure level (ESP), the latter obtained immediately after each individual sang. The sound pressure level is the way to measure the amplitude or loudness of a sound in decibels (dB) as the ratio between the amplitude of the pressure and a reference pressure value, that is, the variation in the pressure of the atmosphere produced by a sound. So, SSP is caused by the bird’s song, and ESP is caused by the set of sounds from natural and human sources when the bird is silent. To do this, two researchers got as close as possible to singing individuals and placed themselves under the bird’s perches (trees, human buildings, electricity poles or cables). The greater the distance at which song and SSP recordings are obtained from the focal bird, the greater the possible negative effects of background noise [51], so we obtained recordings only in the range of 1.5 to 2.5 m between the focal birds and the recording devices.

We started recording a continuous one-hour track (MP3 format, 16 bits, 44.1 kHz) when we approached the first detected individual singing with a SONY ICD-P630F digital recorder (recording frequency range 260–6800 Hz) and a SENNHEISER ME66 + K6 shotgun microphone. One of the two researchers pointed the microphone in the direction of the individual’s beak, recording all the songs until it flew away or stopped singing for a minute. Subsequently, without stopping the recording, we looked for another individual to repeat the procedure until completing the one-hour period. The individuals studied were not marked, so we avoided registering the same individual more than once as far as possible by controlling the initial positions and movements of the birds. For the same reason, we did not know if we recorded the same individuals in the two periods of the day on each one-hectare plot.

At the same time as recording the songs, the second researcher determined the SSP for each song of the focal individual as the maximum value recorded during the song with a PCE-999 sound level indicator (0.1 dB resolution, ±1.5 dB accuracy, measurement ranges 30–130 dB, 31.5–8000 Hz, omnidirectional microphone). To do this, the observer extended the arm as much as possible, pointing the microphone of the sound level indicator at the beak of the individual at an angle of 90° with respect to the longitudinal axis of the bird, without interference from any object, which, together with the short distance, reduces interference from background noise in SSP measurements [52]. After recording SSP for each song, we took an environmental sound pressure level (ESP) measurement when the bird was silent, pointing the sound indicator microphone at the opposite side of the bird for five seconds, recording the maximum observed value as a proxy of the environmental noise present when the individual sang [51]. If the focal individual song interrupted this measurement, we deleted that data. We link each SSP and ESP value to their respective song record, considering the temporal position in the soundtrack. We obtained all sound pressure level measurements with type A frequency weighting (cutting off the lowest and highest frequencies that the average person cannot hear); therefore, the sound pressure level values are expressed as dBA. We assume that our short recording distances produced small effects on the SSP recordings as well as a very small effect of ESP [53]. For that reason, we did not subtract the ESP values from those of SSP [51], allowing us to maintain independent SSP and ESP for further analysis. Movements of the individual’s head or body while singing can also produce variations in the SSP, even over short distances [54], which are difficult to control. Eliminating records in which movements were detected could lead to a significant reduction of the sample; therefore, we assume that the random nature of such events is not aligned with the study factors, such that they could not potentially affect the study results.

We used the set of ESP recorded during the one-hour period as descriptors of the ambient noise present at the time the individuals were singing. The complete description of the soundscape at a particular site is complex in relation to the wide variations in time and space, even at small scales [55]. However, our scope is not a complete description of the soundscape of each plot. Additionally, because each ESP was linked to each song, we used them to explore their relationship with the sound parameters of the songs.

### 2.4. Analysis of Sound Parameters of the Song

Before song processing, we converted MP3 files to WAVE files with Adobe Audition v. 3.0 software. The effects of decompressing MP3 files into WAVE files to obtain song frequency variables from sonograms were evaluated by [56]. The authors found no bias in the frequency parameters but rather a comparatively reduced precision. This low precision appears to have little effect in song frequency comparisons between different species [57], in intraspecific comparisons [58], or when both types of records (MP3 and WAVE) were used together [59,60]. After that, we selected for analysis only clear recordings of songs without other masking sounds, for which their respective SSP and ESP measurements were taken. Because the songs of the Chiguanco Thrush do not have a constant structure (syllables or phrases are not distinguishable) and are highly variable in the sequence and number of different motif and twitter elements (Figure 2), we discretize each song, considering a pause of at least five seconds between them. We cut each selected song and saved them in individual files named in such a way that the observer who obtained the song parameters (N.B.) could not recognize the plot and day period of each song (blind analysis). The total number of songs selected ranged from 16 to 57 per track (Appendix A).

We obtained unfiltered sonograms with Raven Pro v 1.5 software, using Hann’s window, a DFT size of 512, and 50% overlap. From each song, we obtained five parameters: length in seconds, minimum and maximum frequency (manually from the sonograms by placing the cursor on the screen), frequency range (difference between maximum and minimum), and dominant or peak frequency (frequency at the maximum amplitude in the spectrum with the corresponding software tool). It has been suggested that manual methods for obtaining minimum and maximum frequencies from sonograms could produce bias in relation to the observer’s expectation [61] and in relation to the background noise [62], recommending that it is better to obtain measurements from the power spectrum analysis using amplitude thresholds. However, the latter method is also prone to compromising the correct detection of maximum frequencies [16], which also affects measurements of the song’s frequency range [63]. We assume that in our study, the absence of observer expectations and the selection of the sonograms with minimal overlap of background noise (both anthropogenic and biogenic) reduced the possibility of obtaining biased results when obtaining measurements of minimum and maximum frequencies. For further analysis, we grouped all songs by track without regard to the identity of the individuals.

To obtain the number of songs per individual per hour, we counted the total number of songs listened to in each one-hour soundtrack as a proxy for the total number of songs in each dawn and dusk chorus per plot, and then calculated the proportion relative to the number of individuals who sang in that period. The number of individuals recorded in each one-hectare plot during the recording period was 12 to 18 (Appendix A).

### 2.5. Data Analysis

The sound variables of the songs and the environment did not have a normal distribution, so we used a nested general lineal model with a Gamma probability distribution and a logarithmic link function. The model included the factors “zone” (urban and extra-urban), “time of day” (morning dawn chorus and afternoon dusk chorus), and “plot” nested in zone (three plots per zone), using the model:Variable ~ zone + time + zone*time + plot (zone).

We applied a non-parametric Spearman’s rank correlation between the song variables and the ESP, independently for urban and extra-urban zones, to explore the relationships of the song variables between them and evaluate whether they are an immediate response to the ESP. The “number of songs per individual per hour” has normal distribution and homogeneity of variance, so we applied a two-factor general lineal model using “zone” (urban and extra-urban) and “time of day” (morning and afternoon) as factors. We performed all the statistical analysis and graphs with IBM SPSS Statistics v 23 software, considering a significance threshold value of 0.05.

## 3. Results

The environmental sound pressure level (ESP) recorded during the fieldwork confirms that the plots in the urban zone were noisier than the plots in the extra-urban zone, with the afternoons being noisier than mornings in both; this difference was greater in the urban zone, a situation that explains the significant interaction (Table 1 and Table 2 Figure 3A). There was also an effect of the plots within each zone, but without masking the main effect of the zone (Table 2).

Urban Chiguanco Thrush individuals significantly increased their song sound pressure level (SSP) compared to their extra-urban counterparts (Table 1 and Table 2), reaching the highest mean ESP value observed (Figure 3B). The mean SSP of the plots was also different within each zone, but without masking the effect of the zone (Table 2). The SSP values were significantly lower in the afternoons than in the mornings, inversely to the temporal variation of the ESP (Table 1 and Table 2, Figure 3), so the temporal increase in the SSP seems not to be a response to higher levels of ESP, but rather it would respond to other factors not evaluated in this study. In addition to this, each SSP value is not an immediate response to the ESP values, as there was no positive correlation between them in urban and extra-urban zones (Appendix A).

Song length did not differ between urban and extra-urban individuals but was significantly longer in the mornings in the two zones (Table 1 and Table 2, Figure 4A). The number of songs transmitted per individual during a constant period of time was significantly higher in the extra-urban zone (*F*_1,8_ = 53.839, *p* < 0.001) and more intense in the mornings (*F*_1,8_ = 34.681, *p* < 0.001). The significant interaction (*F*_1,8_ = 8.473, *p* = 0.020) explains that the variation between periods in the extra-urban zone was greater than in the urban one (Figure 4B). Thus, Chiguanco Thrush individuals in extra-urban plots produced more and longer songs in the dawn chorus than in the dusk chorus, and urban individuals followed the same basic pattern but reduced the number of songs produced per individual to less than half (Table 1, Figure 4).Urban Chiguanco Thrush individuals showed significantly higher values in all song frequency variables compared to their extra-urban counterparts (Table 2 and Table 3, Figure 3); that is, the entire song moved upward with an average increase of 260 Hz in the dominant frequency. The frequency range increased because the increase in maximum frequencies was proportionately greater than the increase in minimum frequencies (Table 3). It is interesting that in the natural conditions of the extra-urban plots, there was a temporal variation of the frequency parameters, with the exception of the dominant frequency. This variation implies an increase in the frequency range in the morning due to an increase in the maximum frequency and a decrease in the minimum frequency; this general pattern is also present, although attenuated, in the urban plots (Table 2 and Table 3, Figure 3C). This frequency shift response is significantly correlated with the SSP for urban and extra-urban zones (Appendix A); therefore, the mechanism involved in temporarily increasing the frequency range—keeping the dominant frequency relatively constant—is related to the increase in SSP in natural conditions. The inverse relationship with ESP in the extra-urban zone (Appendix A) appears to be a byproduct of the emphasized inverse temporal relationship between ESP and SSP.

## 4. Discussion

### 4.1. There Are Natural Variations in Song Parameters between Dawn and Dusk Choruses in the Chiguanco Thrush

Our results support the first hypothesis we postulated: the songs in the dawn chorus were louder and had larger bandwidths (lower minimum frequency plus higher maximum frequency, with constant peak frequency) compared to the songs in the dusk chorus. As far as we know, variation in structural song parameters between dawn and dusk choruses has not been reported before for any other bird species and may help to better understand the differential functions of dawn and dusk choruses. It is interesting that for extra-urban populations, in which we assume most natural conditions, the variations between dawn and dusk choruses were greater than in urban populations and could imply the restrictions imposed by urban noise, as we will discuss later. In addition to that, our results show that the dawn chorus of both urban and extra-urban populations of Chiguanco Thrush had more and longer songs compared to the dusk chorus. This pattern was widely reported for other bird species, and it is what precisely defines the dawn chorus as the most intense period of acoustic communication throughout a day [31,64].

Such natural temporal adjustments appear not to be a response to temporal variations in environmental sound pressure, first because of the environmental sound pressure level values in extra-urban plots are very low compared to the song sound pressure levels, and second because the inverse relationship that exists between both. Therefore, the production of louder songs with a higher maximum frequency at dawn compared to dusk must respond to the ecological factors responsible for their adaptive value [31,33,64,65]. These structural variations, therefore, would ensure the successful transmission of messages in each of the two choruses, in which, probably, different ecological information is transmitted [66].

One of the general patterns postulated for the dawn chorus is that qualitatively different signals are used compared to the other periods of the day [64]. Evidence with Ovenbirds (*Seiurus aurocapilla*) shows that the same individuals have sufficient plasticity to use different song types between dawn and dusk choruses [66]. Some studies on the genus *Turdus* address these mechanisms of rapid temporal adjustment, which are related to changes in the use of elements of the repertoire. [67] reported for the Eurasian Blackbird (*Turdus merula*) that dawn songs had fewer and shorter motif elements, and [46] reported for the same species the use of songs with a higher proportion of twitter elements when the number of potential competitors increases, which is an important motivation factor [64]. Probably the differential use of repertoire elements in the songs of Chiguanco Thrush is responsible for the modification of the structural parameters of the song. It would be interesting to evaluate in the future the relative proportion of motif and twitter elements used by the Chiguanco Thrush between the dawn and dusk choruses.

### 4.2. The Chiguanco Thrushes Respond to Urban Noise by Increasing the Frequency and Amplitude of Their Songs in Dawn and Dusk Choruses

Our results also support our second hypothesis. Urban Chiguanco Thrushes significantly increased all song frequency parameters, as well as song sound pressure level, in both the dawn and dusk choruses, compared to values in extra-urban habitats. It is important to highlight that within the city, the song sound pressure level increased in both choruses enough to surpass the average urban noise. The combined increase in peak frequency and song sound pressure level in urban settlements has been previously reported for several bird species [17,23,24], including species of the genus *Turdus* [15].

The increase in some song frequency parameters in urban environments is a widely reported response for several bird species around the world (e.g., [14,18,20,21]), but the increase in minimum frequency rather than entire song was mainly reported [19]. Previous studies with species of the genus *Turdus* reported an increase in one or more song frequency parameters in response to urbanization [19,25,36,37,38,39,40]. To our knowledge, this is the first study for the genus *Turdus* to report that the frequency of the entire song increased.

The combined increase in song amplitude and frequency in response to increased environmental sound pressure level is known as the “Lombard effect”, a mechanism shared by all bird species despite their different learning abilities [68]. The dominant frequency and amplitude would increase together when the respiratory system increased the airflow rate. It was postulated that the higher-pitched songs in urban birds could actually be a byproduct of the increase in song amplitude, assuming that the latter is more important for the response to urban noise [23,68,69,70]; but see [71].

However, some evidence in our results with the Chiguanco Thrush indicates that the increase in frequency is important on its own, complementary to the increase in amplitude for the response to urban noise, as previously postulated [24], but see [54,72]. First, we have no evidence for a short-term increase in song sound pressure level in response to increased environmental sound pressure level, which is what the Lombard effect actually proposes (Figure 3A,B and Appendix A), and second, the spectral position of the dominant frequency in urban and extra-urban songs is independent of changes in the song sound pressure level (Appendix A). The increase in song frequency is probably best explained, as we discussed for daily variations, by the decreased use of low-frequency motifs, a pattern also reported for the Eurasian Blackbird [15] and the Song Thrush (*Turdus philomelos*) [25]. In relation to urbanization, there are no reports of this for the genus *Turdus*, but for the Northern Mockingbird (*Mimus polyglottos*), it was reported that the differential use of repertoire elements allows urban individuals to increase the frequency of songs in urban settlements [73]. It would be interesting to explore in the future whether the differential use of repertoire elements is responsible for the response to urbanization by the Chiguanco Thrush.

Furthermore, our evidence shows that extra-urban populations of Chiguanco Thrush have sufficient plasticity to modify their songs between the dawn and dusk choruses. Such abilities could predispose to successful colonization of urban settlements, a proposal in line with what was previously postulated for the same species [35]. In fact, song learning and plasticity [74] and larger bandwidths in songs [75] appear to be better pre-adaptations for colonizing cities. The repertoire of the Chiguanco Thrush is composed of a wide diversity of motif and twitter elements, and, considering all possible combinations and improvisations, this situation potentially provides the previous natural ability to produce highly variable inter- and intra-individual songs. This versatility was also described for other *Turdus* species [25,39,46,67,76,77,78], and could be related to learning and memory abilities [76,79]. Such capabilities would allow a song to respond to different ecological situations in their natural habitats as well as to different habitat alterations by humans, e.g., [80].

### 4.3. Acoustic Communication Responses to Urbanization Are Likely to Represent Serious Tradeoffs for Urban Chiguanco Thrush Individuals

The “noise filter hypothesis” postulates that species that naturally sing at lower frequencies are more sensitive to urban noise, affecting the composition of urban bird communities [8,75,81,82]; but see [83,84]. Therefore, the Chiguanco Thrush could be more pressured in the urban colonization process because extra-urban individuals had a mean minimum frequency around the upper limit of the dominant anthropogenic sound frequencies (2 kHz) [20], and a mean dominant frequency around 650 Hz up. The same could be true for other species of the genus *Turdus*, which also have a minimum frequency of around 2 kHz [25,39,40,85].

In response to the pressure of urban noise on their songs, the urban Chiguanco Thrushes, in addition to the increase in the song sound pressure level, increased the average value of the dominant or peak frequency and increased the frequency of the entire song. Considering that increasing song amplitude is energetically costly [86], urban individuals are probably increasing amplitude to a physiological maximum that restricts them from producing more songs at a time, preventing them from switching songs widely between dawn and dusk choruses, compared to what extra-urban individuals can do. In fact, although urban individuals had songs of the same length as their extra-urban counterparts, they produced less than half the songs per individual. For the Eurasian Blackbird, ref. [87] showed that the energetic state of males determines the timing and intensity of dawn and dusk songs. On the other hand, high-pitched songs that are used more within the city potentially degrade faster [85] and are only useful for short-range communication [25]. The effects on the energetic status, biological fitness and effective acoustic communication [17] of the Chiguanco Thrush populations that survive within the city of La Paz are topics that must be evaluated in the future.

We studied only one species of songbird during the dawn and dusk choruses of a single year. But the mechanisms of song structure variation between dawn and dusk choruses, and even the existence of these variations, may be very different for other songbird species. And it is also possible that the response mechanisms that we have described in this study are not constant over time, as suggested by a recent study [88], particularly if we consider species in which the learned component is important. We hope that this study encourages further studies of song variations between dawn and dusk choruses across a broad spectrum of species and in different cities around the world.

## 5. Conclusions

Our results support the two hypotheses we postulated. Between the dawn and dusk choruses of the Chiguanco Thrush in La Paz, both in extra-urban and urban populations, there are important differences. In addition to the longer and more intensely produced dawn songs, the dawn chorus songs were significantly louder and had larger bandwidths (lower minimum frequency, higher maximum frequency, no change in dominant frequency) than the songs of the dusk chorus. Reporting such structural variations of songs across daily song organization is a novelty and appears not to be a response to environmental sound pressure levels but probably a response to the factors that determine the effective transmission of potentially different messages from the two choruses.

Natural variations between dawn and dusk choruses are wider in extra-urban populations compared to urban populations, where, although the same pattern of song variation is present, it is very constrained by the need to respond to the urban noise. The response to the urbanization of the Chiguanco Thrush involves raising the frequency of the entire song to approximately 260 Hz with respect to the songs produced in extra-urban habitats, coupled with increasing the song sound pressure level enough to surpass the sound pressure of the urban noise. Such changes occur in dawn and dusk choruses. These energetically demanding changes likely have as a by-product a reduction in the number of songs produced per hour within the city, highlighting the need to assess whether urban populations are physiologically stressed. Understanding the natural structural variations between the dawn and dusk choruses could allow for a better interpretation of how some bird species cope with urban challenges. We propose that the ability to change song parameters between morning and afternoon allows the Chiguanco Thrush to better respond to the new, noisy urban environment.

## Figures and Tables

**Figure 1 animals-14-00646-f001:**
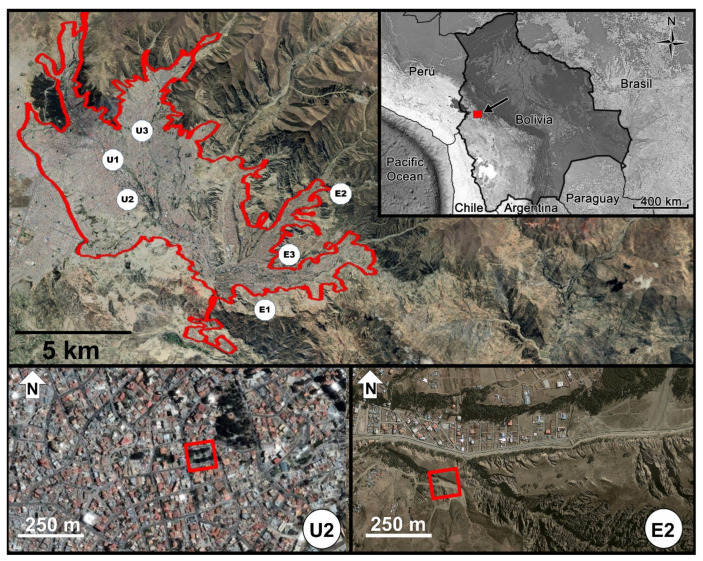
In the upper panel, the city of La Paz with the limits of the urban fringe in November 2013 (red polygon), the upper right square details the position of the study area in the Bolivian Andes (red solid square). The position of the study plots is marked with white circles (U1 to U3 for the urban zone, and E1 to E3 for the extra-urban zone). Two plots of each zone (U2 and E2) are shown in detail in the lower panel as examples of the general landscape. Satellite images from Google Earth Pro 7.3.2, 2019 (11 October 2013).

**Figure 2 animals-14-00646-f002:**
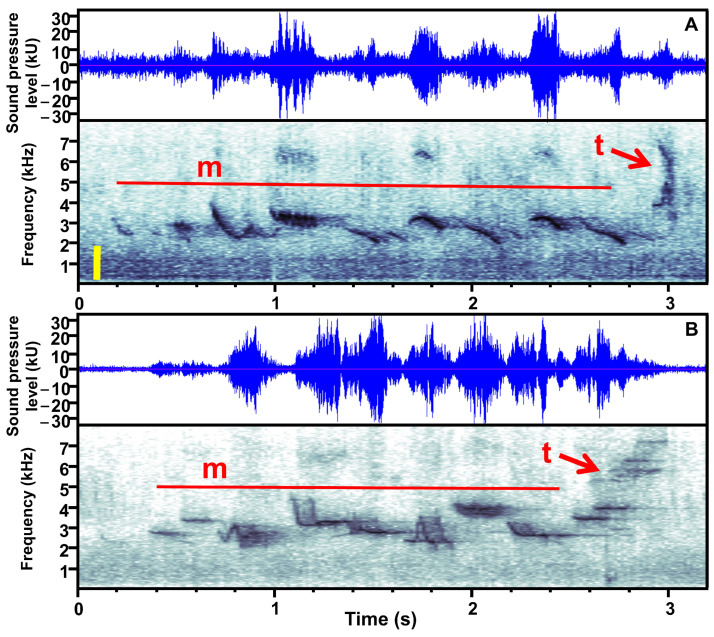
Graphs of waveforms (top) and spectograms (below) as a function of time, for recordings from the U2 urban plot (**A**), and the E1 extra-urban plot (**B**) in La Paz, Bolivia, including a representative song of the Chiguanco Thrush (*Turdus chiguanco*) with the soundscape. In waveforms, kU refers to “kilo Units” of signal sample values that are proportional to the sound pressure level. The spectograms for each bird song show several motif elements (m) and one twitter element (t). The yellow bar in the spectogram of A shows the bandwidth of urban anthropogenic noise. Graphs were created with Raven Pro v. 1.5 (Bioacoustics Research Program, 2011).

**Figure 3 animals-14-00646-f003:**
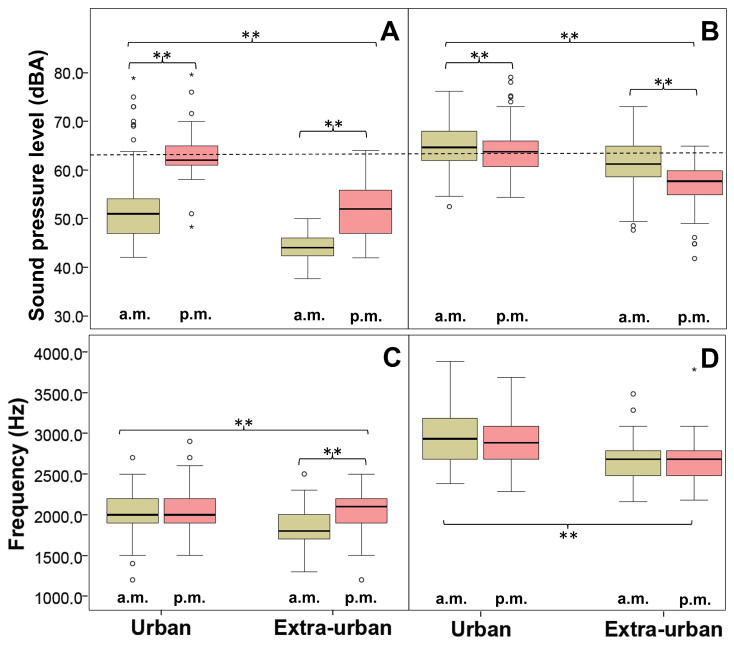
Boxplots for environmental sound pressure level ESP (**A**), and song variables of the Chiguanco Thrush: song sound pressure level SSP (**B**), minimum song frequency (**C**), and dominant song frequency (**D**), for urban and extra-urban zones of the city of La Paz, and two periods of the day: morning (yellow) and afternoon (red). The dotted line in the top panel represents the average ESP value for the urban zone in the afternoon (maximum average ESP detected). The symbol ** indicates significant differences (*p* < 0.001) between groups, and the open circles and single stars indicate outliers.

**Figure 4 animals-14-00646-f004:**
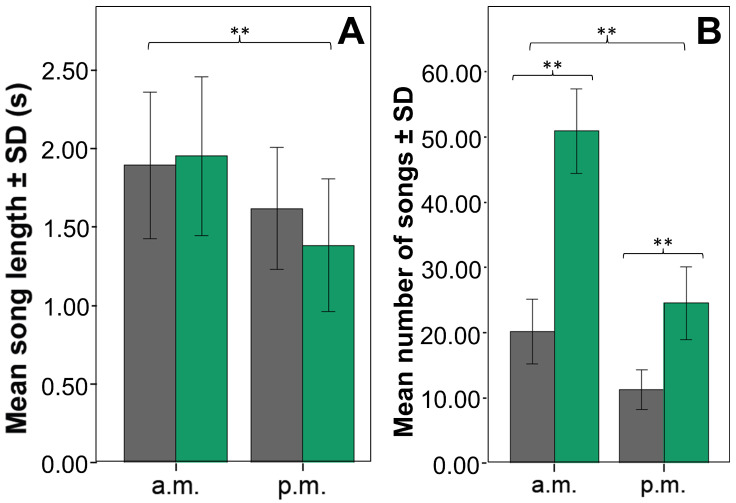
Bars for the mean song length (**A**) and for the mean number of songs emitted per individual in one-hour (**B**) for Chiguanco Thrush populations in urban (grey bars) and extra-urban (green bars) zones of the city of La Paz, in two periods of the day, morning (a.m.) and afternoon (p.m.). The symbol ** indicates significant differences (*p* < 0.001) between groups.

**Table 1 animals-14-00646-t001:** Means and deviations of the acoustic variables of the environment and songs of the Chiguanco Thrush in the city of La Paz, separated between urban (U) and extra-urban (E-u) zones, and for two periods of the day, morning (a.m.) and afternoon (p.m.). SSP is the song sound pressure level, ESP is the environmental sound pressure level and N is the total number of songs (for song length, SSP and ESP), or the total number of complete sound tracks analyzed to determine the number of songs per individual per hour.

		N	Song Length (s)	SSP (dBA)	ESP (dBA)	N	Songs Number
**U**	**a.m.**	113	1.89 ± 0.47	64.70 ± 4.49	52.26 ± 7.34	3	20.18 ± 5.01
	**p.m.**	86	1.62 ± 0.39	64.03 ± 4.95	63.01 ± 4.26	3	11.25 ± 3.07
**E-u**	**a.m.**	162	1.95 ± 0.51	61.75 ± 4.80	44.17 ± 2.42	3	50.91 ± 6.52
	**p.m.**	73	1.38 ± 0.42	56.70 ± 4.51	51.45 ± 5.72	3	24.52 ± 5.56

**Table 2 animals-14-00646-t002:** Results of the General Lineal Model test using Gamma distribution and logarithmic link function, for the variables of the Chiguanco Thrush song and an environmental variable (environmental sound pressure level). The Wald’s value test is included in each cell with the *p*-value in parenthesis. The factor “zone” has two levels (urban and extra-urban), the factor “time” has two levels (morning and afternoon), and the factor “plot” has three levels nested at each zone level. The model was: variable ~ zone + time + zone*time + plot (zone).

	Intercept	Zone	Time	Zone*Time	Plot (Zone)
**Minimum** **frequency**	1,419,631.481 (<0.001)	24.626 (<0.001)	23.483 (<0.001)	11.390 (=0.001)	60.804 (<0.001)
**Maximum frequency**	325,706.139 (<0.001)	9.795 (=0.002)	19.175 (<0.001)	2.461 (=0.117)	3.718 (=0.445)
**Frequency range**	104,510.041 (<0.001)	5.619 (=0.018)	23.450 (<0.001)	4.005 (=0.045)	4.757 (=0.313)
**Dominant frequency**	1,650,781.477 (<0.001)	58.045 (<0.001)	0.938 (=0.333)	1.517 (=0.218)	11.712 (=0.020)
**Song sound pressure level**	1,354,249.158 (<0.001)	139.820 (<0.001)	49.586 (<0.001)	32.400 (<0.001)	90.647 (<0.001)
**Environmental sound pressure level**	775,722.214 (<0.001)	450.657 (<0.001)	353.687 (<0.001)	4.299 (=0.038)	55.149 (<0.001)
**Song length**	1630.271 (<0.001)	2.445 (=0.118)	95.284 (<0.001)	13.342 (<0.001)	37.315 (<0.001)

**Table 3 animals-14-00646-t003:** Means and deviations of variables related to frequency of the songs of the Chiguanco Thrush, separated between urban (U) and extra-urban (E-u) zones of La Paz city, and for two periods of the day, morning (a.m.) and afternoon (p.m.). N is the total number of songs analyzed.

		N	Minimum (Hz)	Maximum (Hz)	Range (Hz)	Dominant (Hz)
**U**	**a.m.**	113	2018.58 ± 266.11	6028.76 ± 1569.94	4010.18 ± 1615.72	2963.70 ± 315.83
	**p.m.**	86	2058.14 ± 287.47	5582.56 ± 1542.82	3524.42 ± 1593.01	2881.11 ± 320.38
**E-u**	**a.m.**	162	1843.21 ± 241.41	5765.12 ± 1688.91	3921.91 ± 1739.82	2655.14 ± 217.89
	**p.m.**	73	2049.31 ± 245.59	4835.62 ± 1634.30	2786.30 ± 1644.97	2671.74 ± 292.62

## Data Availability

The raw data supporting the conclusions of this article will be made available by the authors upon request.

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
