# Peer review of "Natural Patterns in the Dawn and Dusk Choruses of a Neotropical Songbird in Relation to an Urban Sound Environment"

_animals, 2024, doi:10.3390/ani14040646_

Round 1

Reviewer 1 Report

Comments and Suggestions for Authors

Dear Authors,

Thank you for your valuable research work as a contribution to knowledge about the influence of urbanization on birdsong.

Apart from minor remarks I have major remarks related to the methodology and scientific reporting.

·         Textual: lines 20, 37, 93

·         About the title: ‘… dawn and dusk singing …’ could be used instead of ‘… choruses…’. Chorus has in general a multi-species character, which is not included in this work, as it is focusing on a single bird.

·         Regarding the measurements of SSP and ESP: a PCE-999 sound level indicator was used. This device uses a frequency rating, A or C, so the dB values are indicated as dBA or dBC. The A weighting was used in this experiment, so the values can be reported as dBA values as the ‘sound pressure levels’ and not the ‘sound pressures’.

·         Related tot he measurement of the ‘SSP’. The idea is to measure the source level of the bird as a sound producer. In doing such measurements the distance to the source is crucial. Only recordings with distances in the range of 1.5 to 2.5 m have been considered. But if one considers an omnidirectional source a measurement at 1.5 m or at 2.5 m will strongly differ. This raises questions about the validity of these measurements. An approach could be to measure at different places simultaneously to estimate the (bird) source level. Distances will probably also vary with the typical vegetation of a study plot. (Aside, I’m also surprised to be able to measure at such a close distance to the bird.)

·         Directivity issues: Is there information about the directivity of the PCE-999? I would assume it is quite omnidirectional (which eliminates the need to point at the bird). On the contrary, the bird as a source has some directivity (some references about this could be discussed). This should be considered, in the end to only use recordings with a similar orientation towards the bird.

·         Figure 2: On top the signal itself is shown, and not the amplitude (envelope).

·         Considering the data analysis a nested GLM was used to understand the variability in the sound variables. In fact, the model uses only categorical inputs. It is not clear how these are numerically taken into the model. The model is a nested model which is typical for data with some hierarchy. What type of hierarchy/nesting was used?

·         In the tables, please add units to the values reported. (Are they log related?)

Comments on the Quality of English Language

E.g. lines 20, 37, 93

Author Response

R: We are very grateful for the positive comments on our manuscript, as well as the observations on the conceptual background of our manuscript. We have responded to all comments, which, we hope, has allowed us to substantially improve the understanding of our manuscript.

  • Textual: lines 20, 37, 93

R: Line 20: We have rewritten the simple summary in order to make it clearer for the general public while staying within the maximum number of words requested.

Line 37: we have reviewed and modified the keywords

Line 93: The phrase was rewritten to "with no more than a million inhabitants "

  • About the title: ‘… dawn and dusk singing …’ could be used instead of ‘… choruses…’. Chorus has in general a multi-species character, which is not included in this work, as it is focusing on a single bird.

R: Actually, the definition of animal chorus is the set of vocalizations emitted by multiple individuals of the same or different species at the same time (Burt & Vehrencamp 2005, Catchpole & Slater 2008, Marín-Gómez & MacGregor-Fors 2021,and in line with this, several articles studies a chorus of a particular species (e. g. Mace 1987, Cuthill & Macdonald 1990, Barnett & Briskie 2007, Dalziell & Cockburn 2008, Quiroz-Oliva & Sosa-López 2022). However, considering the different interpretations of the concept, and in order to be as broad as possible for all readers, we have changed the title and the context in the manuscript to avoid saying that "they are the choirs of this species", and instead say that "the species participates in these choruses”. The new title is: Natural differences in a Neotropical songbird's songs between dawn and dusk choruses suggest mechanisms for responding to urbanization.

Barnett, C.A., Briskie, J.V. 2007. Energetic state and the performance of dawn chorus in silvereyes (Zosterops lateralis). Behav Ecol Sociobiol 61, 579–587.

Burt, J.M., Vehrencamp, S.L. 2005. Dawn chorus as an interactive communication network. In McGregor, P.K. (ed) Animal Communication Networks: 320–343. Cambridge: Cambridge University Press.

Catchpole, C.K., Slater, P.J.B. 2008. Bird Song: Biological Themes and Variations. New York: Cambridge University Press.

Dalziell, A. H., Cockburn, A. 2008. Dawn song in superb fairy-wrens: a bird that seeks extrapair copulations during the dawn chorus. Animal Behaviour, 75, 489-500.

Mace, R. 1987. The dawn chorus in the great tit Parus major is directly related to female fertility. Nature 330, 745–746.

Cuthill, I.C., Macdonald, W.A. 1990. Experimental manipulation of the dawn and dusk chorus in the blackbird Turdus merula . Behav Ecol Sociobiol 26, 209–216.

Marín‐Gómez, O. H., MacGregor‐Fors, I. 2021. A global synthesis of the impacts of urbanization on bird dawn choruses. Ibis, 163, 1133-1154. doi:10.1111/ibi.12949.

Quiroz-Oliva, M., Sosa-López, J.R. 2022. Vocal behaviour of Sclater’s Wrens, a duetting Neotropical songbird: repertoires, dawn chorus variation, and song sharing. J Ornithol 163, 121–136.

  • Regarding the measurements of SSP and ESP: a PCE-999 sound level indicator was used. This device uses a frequency rating, A or C, so the dB values are indicated as dBA or dBC. The A weighting was used in this experiment, so the values can be reported as dBA values as the ‘sound pressure levels’ and not the ‘sound pressures’.

R: Thank you very much for pointing out, you are right. We change throughout the entire article dBA instead of only dB.

  • Related tot he measurement of the ‘SSP’. The idea is to measure the source level of the bird as a sound producer. In doing such measurements the distance to the source is crucial. Only recordings with distances in the range of 1.5 to 2.5 m have been considered. But if one considers an omnidirectional source a measurement at 1.5 m or at 2.5 m will strongly differ. This raises questions about the validity of these measurements. An approach could be to measure at different places simultaneously to estimate the (bird) source level. Distances will probably also vary with the typical vegetation of a study plot. (Aside, I’m also surprised to be able to measure at such a close distance to the bird.)

R: We have been aware of the effect of the distance of the bird from the pressure level indicator microphone throughout the study. That is why we have made an enormous effort to record as close as possible to the individuals, always positioning ourselves under their perch and extending our arm as far as possible. The vast majority of individuals recorded in urban and extra-urban environments were singing on perches approximately 4.3 m above the ground (with variations in the range of 50 cm above and 50 cm below). The need to record them as quickly as possible made it impossible to make exact distance measurements without the birds escaping. We know that these distances may imply variations in sound pressure measurements, however we have carried out exactly the same method in all circumstances, whether spatial or temporal, so that if biases exist, they would exist for all measurements without aligning with any study factor. For this reason, we are convinced that our results allow us to reach the proposed conclusions.

  • Directivity issues: Is there information about the directivity of the PCE-999? I would assume it is quite omnidirectional (which eliminates the need to point at the bird). On the contrary, the bird as a source has some directivity (some references about this could be discussed). This should be considered, in the end to only use recordings with a similar orientation towards the bird.

R: Yes, microphone of the PCE-999 sound level indicator is omnidirectional, as it meets the standard ANSI.s.1, we are including this specification in the text. It would certainly not be necessary to aim at the bird's beak, but by stretching the arm as far as possible in the direction of the bird that is singing below the observers, the direction and position was as we described and we have done this procedure in all cases in order to be consistent.

  • Figure 2: On top the signal itself is shown, and not the amplitude (envelope).

R: You are right. We have corrected its identification and units on the "y" axis, which are actually kilounits (kU) according to the output of the software used. We have clarified all of this in the new wording of the legend: “Figure 2 . Graphs of waveforms (top) and spectograms (below) as a function of time, for recordings from the U2 urban plot (A), and the E1 extra-urban plot (B) in La Paz, Bolivia, including a representative song of the Chiguanco Thrush (Turdus chiguanco) with the soundscape. In wave-forms, kU refers to “kilo Units” of signal sample values that are proportional to the sound pressure level. The spectograms for each bird song show several motif elements (m) and one twitter element (t). The yellow bar in the spectogram of A shows the bandwidth of urban anthropogenic noise. Graphs were created with Raven Pro v. 1.5 (Bioacoustics Research Program, 2011)..”

  • Considering the data analysis a nested GLM was used to understand the variability in the sound variables. In fact, the model uses only categorical inputs. It is not clear how these are numerically taken into the model. The model is a nested model which is typical for data with some hierarchy. What type of hierarchy/nesting was used?

R: The gamma function is defined for all complex numbers except non-positive integers, so it is used to find relationships between a continuous (non-categorical) dependent variable - like ours - with one or more continuous or categorical predictive variables. We selected Gamma probability distribution with logarithmic link function because our variables were strongly skewed to higher positive values, so we introduced our dependent variables without transformation. GLM allows introducing predictive categorical variables into the model, in our case zone (two levels), time (two levels) and plot (three levels). We explain that in methods, and in there we specify the model in which the plots are nested in zone, that means, we look for the effect that the three different plots could have within each of the zones (urban or extra-urban) to which they belong.

  • In the tables, please add units to the values reported. (Are they log related?)

R: You are right, in table 2 we included Hz as units. Variables were not log transformed, as we explain in methods (Data analysis) as well in the previous answer.

Reviewer 2 Report

Comments and Suggestions for Authors

Comments on animals-2816722
Title: Natural differences between the dawn and dusk choruses of a Neotropical Songbird and their relationship to its response to urbanization

This study tried to compare the natural differences between the dawn and dusk choruses of a Neotropical Songbird, since no studies have been conducted tocompare such potential changes between dawn and dusk choruses.

I do like this idea, and think it well written.

My major concern about this study is that how the authors control for individual bird song for analysis, as they did not band any bird individual, and did not make sure if they recorded the same individuals in the two periods of the day.

For example, if they use the dawn chorus from bird A but use dusk chorus from bird B, it would not be the natural differences between the dawn and dusk choruses of a bird.

In addition, in the Results I could not find the sample size; they should state clearly how many bird individuals they recorded and used for later analysis.
Thus I suggest they should put Table 1 and Table 2 in the text, not as
SUPPLEMENTARY MATERIAL.

Author Response

We are very grateful for the reviewer's positive assessment of our manuscript. We also greatly appreciate the specific comments, which allow us to visualize some elements that we did not explain well before and created confusion.

  • My major concern about this study is that how the authors control for individual bird song for analysis, as they did not band any bird individual, and did not make sure if they recorded the same individuals in the two periods of the day. For example, if they use the dawn chorus from bird A but use dusk chorus from bird B, it would not be the natural differences between the dawn and dusk choruses of a bird.

R: The observation is very interesting. However, the objective of the study was not to control the songs emitted by each individual, but rather to have a random sample of songs from several individuals from each population in each plot. In our next response we explain the reason for the confusion we possibly created in sentence 233-235. For this same reason, it was not the objective to record the same individuals in the two periods of the day in each plot and compare the songs of each individual in the dawn and dusk choruses. This would serve for a study of characterizations of inter-individual and intra-individual variations within each population, and personality studies, which was not the objective of this study.

  • In addition, in the Results I could not find the sample size; they should state clearly how many bird individuals they recorded and used for later analysis. Thus I suggest they should put Table 1 and Table 2 in the text, not as SUPPLEMENTARY MATERIAL.

R: We understand the reviewer's proposal. In part this arises from a previous wording in the sentence in question (lines 233-235), which seemed to say that we analyzed a certain number of songs from a certain number of individuals. In reality, we work only with the songs with the best signal and complete SSP and ESP data in the one-hour soundtrack, therefore, it is a random sample of songs from each population sampled for a certain period. We have modified this sentence to “After that, we selected for analysis only clear recordings of songs without other masking sounds and for which their respective SSP and ESP measurements were taken”. Therefore, the sample for the acoustic analyzes is not the total number of individuals recorded in the soundtrack (which appears in table 1 of Supplementary Material and allows to calculate the number of songs/individual/hour) but the number of songs analyzed (Tables 1 and 2). We believe that moving the tables currently in Supplementary Material to the main text could overload the manuscript and we would prefer to keep them there as well because none of the other four reviewers observed this. However, we will also consult the editor's opinion, and if he agrees that moving these tables would be best without affecting the understanding of the main text, we will do so.

Reviewer 3 Report

Comments and Suggestions for Authors

Summary: This paper studies the singing of urban and rural populations of the Chiguanco Thrush and compares several characteristics of the species' song between populations.  Uniquely, comparisons are also made between dawn and dusk choruses.  Natural variations between dawn and dusk were that dawn songs were louder and with larger bandwidths.  Adaptation to urban noise is demonstrated with a rise in frequency of song and an increase in amplitude.

General Comments: Paper as a whole is well structured, clearly written and relevant to the journal.  The reference list is extensive with a mixture of classic papers and more recent research particularly focussed on urban ecosystems.

Introduction is concise and generally well written - some specific minor comments below.  Choice of study species is well justified, sensible to choose from a geographically widespread genus.  Hypotheses are clearly stated and  the experiments conducted were appropriate to test these hypotheses.    Enough information and detail presented in the methods to give clear understanding of data collection and data analysis. Some minor clarification required in the methods - see comments below.

Results are described concisely in the text and suitably displayed in tables and figures (including information in supplementary material).  Suitable analysis is conducted and sample sizes are large enough/data robust enough for conclusions to be drawn.  Discussion is thorough and the results are significantly and appropriately interpreted.  

Overall this is a well conducted, well reported and fascinating study which will be of interest to readers of the journal.  There are increasing numbers of studies on birds and bird song in urban environments, but as the authors state, the comparison between dawn and dusk song parameters has had little previous mention.

Specific comments:

Lines 54/55: a definition of "Sound pressure" would be useful here

Lines 59/60: I'd argue that dawn and dusk choruses are only common in temperate species.  Many tropical species do not display these patterns, so probably only a minority habit.

Lines 82/83: define what is meant by "proportion of songs produced per time"

Lines 142-146: given the context, is it not possible to tell the 2 species apart by their songs?

Lines 178/179: define SSP and ESP

Lines 209/210: No problem with what is being done here, but not sure of the logic of referring to what the average person can hear, rather than what the average thrush can hear...

Line 233: "prone" songs?

Comments on the Quality of English Language

Minor errors of language only, no instances where meaning is unclear.

Author Response

We are very grateful for the positive comments on our manuscript, as well as the observations on the conceptual background of our manuscript. We have responded to all comments, which, we hope, has allowed us to substantially improve the understanding of our manuscript.

  • Specific comments:
  • Lines 54/55: a definition of "Sound pressure" would be useful here

R: As another reviewer suggested, we will henceforth use the more precise concept of "sound pressure level." We consider that including a technical definition of this concept in the introduction would not be appropriate and could generate confusion for the understanding of the paragraph, but we believe that including in parentheses the colloquial word (loudness) will be sufficient for the understanding of the concept by all audiences. We include later, in methods, a more precise definition. In the simple summary, for the general public, we prefer to use “loudness”.

  • Lines 59/60: I'd argue that dawn and dusk choruses are only common in temperate species.  Many tropical species do not display these patterns, so probably only a minority habit.

R: Actually, dawn and dusk choruses are recognized to occur in tropical environments too (Burt & Vehrencamp 2005, Marín-Gómez & MacGregor-Fors 2021), and there are numerous interesting studies in the tropics (e.g. Berg et al. 2006, Marín-Gómez & MacGregor-Fors 2019, Stanley et al 2016). Obviously, the volume of publications for the tropics compared to temperate zones is really small, but precisely that motivates more research in the tropics. Several species in the tropics probably do not follow this temporal pattern, another topic to be studied. In the city of La Paz, the dawn and dusk choruses are truly fantastic and the Chiguanco Thrush one of the main protagonists. We are including a clarification in the fourth introductory paragraph: “Two more intense periods called dawn and dusk choruses are recognized in the daily organization of song production in most birds of temperate and tropical latitudes”.

Berg, K. S., Brumfield, R. T., & Apanius, V. (2006). Phylogenetic and ecological determinants of the neotropical dawn chorus. Proceedings of the Royal Society B: Biological Sciences, 273(1589), 999-1005.

Burt, J.M., Vehrencamp, S.L. 2005. Dawn chorus as an interactive communication network. In McGregor, P.K. (ed) Animal Communication Networks: 320–343. Cambridge: Cambridge University Press.

Marín-Gómez, O. H., & MacGregor-Fors, I. (2019). How early do birds start chirping? Dawn chorus onset and peak times in a Neotropical city. Ardeola, 66(2), 327-341.

Marín‐Gómez, O. H., MacGregor‐Fors, I. (2021) A global synthesis of the impacts of urbanization on bird dawn choruses. Ibis, 163, 1133-1154. doi:10.1111/ibi.12949.

Stanley, C. Q., Walter, M. H., Venkatraman, M. X., & Wilkinson, G. S. (2016). Insect noise avoidance in the dawn chorus of Neotropical birds. Animal Behaviour, 112, 255-265.

  • Lines 82/83: define what is meant by "proportion of songs produced per time"

R: From now on, we have made the concept more precise using: "number of songs per individual".

  • Lines 142-146: given the context, is it not possible to tell the 2 species apart by their songs?

R: The songs of all thrushes have great intra- and inter-individual variation, mainly because they include a high degree of modifications and improvisations, for this reason it is difficult to distinguish these species by song alone. But this species is very rare to observe, it seems that it enters temporarily and/or its populations are very low, so the possibility of confusion was very low, and in the very few cases we observed it, we exclude it through morphology. This paragraph was more aimed at readers knowledgeable about the composition of birds in the city of La Paz, who might question what we did in the event of confusing an individual of the Great Thrush. Another reviewer suggested us to delete this paragraph to avoid unnecessary confusion. Understanding that the article is intended for a broader audience, we agree with this suggestion and have removed this paragraph.

  • Lines 178/179: define SSP and ESP

R: We are including a more complete explanation: “The sound pressure level is the way to measure the amplitude or loudness of a sound in decibels (dB) as the ratio between the amplitude of the pressure and a reference pressure value, that is, the variation in the pressure of the atmosphere produced by a sound. So, SSP is caused by the bird's song and ESP is caused by the set of sounds from natural and hu-man sources when the bird is silent”.

  • Lines 209/210: No problem with what is being done here, but not sure of the logic of referring to what the average person can hear, rather than what the average thrush can hear...

R: Interesting observation, but the devices are standardized for human hearing, and standards for other animal species are not known. However, with the understanding that bird songs can be heard by humans, bioacoustics studies use these standardizations and the most used is type A frequency weighting.

  • Line 233: "prone" songs?

R: We agree, the sentence constructed before did not make clear what we wanted to say and only created confusion, we have changed it to “After that , we selected for analysis only clear recordings of songs without other masking sounds and for which their respective SSP and ESP measurements were taken”

Reviewer 4 Report

Comments and Suggestions for Authors

This is a careful and detailed analysis of the calls of a Turdus species in relation to the urban-exurban interface. It is an observational study with appropriate and testable hypotheses. It is a "pure" study, asking questions about how nature works, rather than solving a problem as an applied study would do. I do these kinds of studies, too, and I think it's important to make such data available in published form.

I only have a few suggestions related to the science of your work:

Lines 142-146: I appreciate the care necessary to ensure you are studying only one species, but I think your reader will assume you knew to be careful. There is not a need to introduce doubt. It seems unfounded to me. I suggest you delete this paragraph.

Figures 3 and 4: First, let me say I loved all your Figures! Excellent captions and display. Everything was easy to understand. However, is there a way to highlight on the Figures the key statistical differences you found? I had to refer back to the text to see if one boxplot or bar was different than another.

Lines 483-485: When you say there is no evidence of Lombardi effect, I believe you. However, it would help if you referred to Figure 3B here and explained what the Lombardi effect would look like if it were present. 

Line 489: I think you mean "daily" and not "dairy"

Line 536: I did not like your final sentences here. The problem is that you did not measure human health, nor bird health, so it is not appropriate to say that your results show it is important to pay more attention to noise pollution. It would be okay if you cited some more literature on these topics, but lacking that, you've stretched the importance of your work too far.

Comments on the Quality of English Language

Title: I find this title awkward because it is not easy to understand the objects of "their" and "its." May I suggest: Natural patterns in the dawn and dusk choruses of a Neotropical songbird in relation to an urban sound environment. 

Line 20: there is an extra word, "to"

Line 22: the word "in" should be "is"

Line 31: using semicolons would help organize your parameters. Do this: "song length; song sound pressure; minimum, maximum, range and dominant frequency; and the proportion of songs produced."

Line 67: see typo

Lines 71-75: split into more than one sentence

Lines 73 and 80: Delete the adjective "hardest," as it is not necessary and I'm not sure what it means anyway.

Line 82: use semicolons as suggested above

Line 93: delete "of"

Line 149: change the word "on" to "during"

Lines 190-202: make sure all these sentences are in past-tense

Line 216: change "imply" to "lead to"

Lines 220-223: split into more than one sentence

Lines 233-235: I don't understand this sentence. Please re-phrase.

At this point, I continued to find minor problems with the English, but I stopped noting them. I recommend another person read this over to make sure your word choices are appropriate and clear.

Author Response

We are very grateful for the positive comments on our manuscript, and especially for the carefully review of the document that has allowed us to realize several errors that we previously made.

  • I only have a few suggestions related to the science of your work:
  • Lines 142-146: I appreciate the care necessary to ensure you are studying only one species, but I think your reader will assume you knew to be careful. There is not a need to introduce doubt. It seems unfounded to me. I suggest you delete this paragraph.

R: The suggestion is important. This paragraph was more aimed at readers knowledgeable about the composition of birds in the city of La Paz, who might question what we did in the event of confusing an individual of the Great Thrush. But this species is very rare to be observed, it seems that it enters temporarily and/or its populations are very low. Understanding that the article is intended for a broader audience, we agree with the reviewer's suggestion, and have removed this paragraph to avoid unnecessary confusion.

  • Figures 3 and 4: First, let me say I loved all your Figures! Excellent captions and display. Everything was easy to understand. However, is there a way to highlight on the Figures the key statistical differences you found? I had to refer back to the text to see if one boxplot or bar was different than another.

R: We greatly appreciate your positive comments about our figures. According to your suggestion, we have incorporated the symbology to indicate the existence of significant differences between groups in Figures 3 and 4, and we have added the respective explanation in the legends.

  • Lines 483-485: When you say there is no evidence of Lombardi effect, I believe you. However, it would help if you referred to Figure 3B here and explained what the Lombardi effect would look like if it were present. 

R: Thank you for your suggestion, we have included reference to Figure 3 and Table 2 of supplementary material in this sentence.

  • Line 489: I think you mean "daily" and not "dairy"

R: You are right, we corrected that.

  • Line 536: I did not like your final sentences here. The problem is that you did not measure human health, nor bird health, so it is not appropriate to say that your results show it is important to pay more attention to noise pollution. It would be okay if you cited some more literature on these topics, but lacking that, you've stretched the importance of your work too far.

R: We agree. We include this paragraph to provide context in the manuscript for the public applied knowledge phrase that is suggested to be present in the simple summary. But certainly, in the manuscript it is not linked to our results. We have removed it.

  • Comments on the Quality of English Language
  • Title: I find this title awkward because it is not easy to understand the objects of "their" and "its." May I suggest: Natural patterns in the dawn and dusk choruses of a Neotropical songbird in relation to an urban sound environment

R: We understand the writing problems, we have changed the title to: “Natural differences in a Neotropical songbird's songs between dawn and dusk choruses suggest mechanisms for responding to urbanization”.

  • Line 20: there is an extra word, "to"

R: Done, actually we have rewritten the simple summary in order to make it clearer for the general public while staying within the maximum number of words requested.

  • Line 22: the word "in" should be "is"

R: Done

  • Line 31: using semicolons would help organize your parameters. Do this: "song length; song sound pressure; minimum, maximum, range and dominant frequency; and the proportion of songs produced."

R: Done

  • Line 67: see typo

R: Done

  • Lines 71-75: split into more than one sentence

R: Done

  • Lines 73 and 80: Delete the adjective "hardest," as it is not necessary and I'm not sure what it means anyway.

R: We have specified the concept to "heavily urbanized areas".

  • Line 82: use semicolons as suggested above

R: Done

  • Line 93: delete "of"

R: In agree, the phrase was rewritten to "with no more than a million inhabitants"

  • Line 149: change the word "on" to "during"

R: Done

  • Lines 190-202: make sure all these sentences are in past-tense

R: We have revised the text and corrected past tense errors.

  • Line 216: change "imply" to "lead to"

R: Done

  • Lines 220-223: split into more than one sentence

R: Done

  • Lines 233-235: I don't understand this sentence. Please re-phrase.

R: We agree, the sentence constructed before did not make clear what we wanted to say and only created confusion, we have changed it to “After that , we selected for analysis only clear recordings of songs without other masking sounds and for which their respective SSP and ESP measurements were taken”

  • At this point, I continued to find minor problems with the English, but I stopped noting them. I recommend another person read this over to make sure your word choices are appropriate and clear.

R: We are very grateful for the previous suggestions and we revised the length of the entire document.

Reviewer 5 Report

Comments and Suggestions for Authors

The authors present an interesting study on the differences between dawn and dusk choruses in a passerine bird and how these are influenced by urbanization. The idea is quite novel and the methods, results and conclusions are generally well presented. I have some minor comments regarding some aspects of the methodology that could be further clarified by the authors.   

Line 16: proportion of what?

Lines 18-19: less than half compared to what?

Line 20: what type of danger?

Line 28: what exactly do you mean with “natural” variation? I think it is better to explicitly state what type of variation you would expect, for example frequency ranges, amplitude differences, song production rates and so on

Line 73: what do you mean “hardest urban core” here and in line 80? Perhaps change with “highly urbanised areas”

Line 83: would be better to state you time unit, for example “songs produced per hour” or “…per minute”

Line 84: the choruses do not “face” the soundscapes, they are rather “produced” or “occur” within certain urban soundscapes

Line 85: still, not clear what you mean with “natural” variation. Of course there will be variation, but here you should clarify what type of variation you expect and why

Line 156-157: were the point counts conducted once per plot? Were they in the center of each plot? And at what time of the day were they conducted? Point counts should be conducted more than once, I would expect at least 3-4 at standard times or at different times during the day depending on the type of study

Line 194: what about territoriality in this species? During the breeding period males of many passerines, including trushes, sings within their territory, sometimes from selected perches. If that is so, you could have just controlled for this by taking GPS points of the recording and avoid recording multiple times in the same spot during subsequent visits

Line 215-218: sure it is unlikely that the singing bird would maintain the same posture during the recording of SSP. However, here rather than “assuming” you could have done earlier preliminary tests. For example, you could use a recording (for example a song from Xenocanto) broadcasted from a speaker at a standard amplitude and change the position of the speaker in relation to the sound level recorder. This would tell you much variation in SSP values you should expect if a bird moves the head in the opposite direction of the recorder.

Line 233-234: what you mean with “prone songs”? Before you said that it was impossible to know if you recorded the same individuals more than once, so how can you distinguish a song from a focal individual? I think here you mean only that you selected clear recordings of songs with no other background noise and with SSP and ESP measurements taken.

Table 2: please write the measurement units for the frequency values.

Line 489: change “diary” with “daily”

Comments on the Quality of English Language

English language is fine, only minor editing necessary

Author Response

We are very grateful for the reviewer's positive comments. we are also very grateful for the helpful comments and suggestions, we considered them all and greatly improved the clarity of our manuscript.

  • Line 16: proportion of what?

R: From now on we will be clearer with this definition and use “number of songs per individual per hour” in methods we have established a better definition: “To obtain the number of songs per individual per hour, we counted the total number of songs listened to in each one-hour soundtrack, as a proxy for the total number of songs in each dawn and dusk chorus per plot, and then calculated the proportion relative to the number of individuals who sang in that period”.

  • Lines 18-19: less than half compared to what?

R: We have clarified that it is in relation to non-urban individuals

  • Line 20: what type of danger?

R: We have rewritten the simple summary in order to make it clearer for the general public while staying within the maximum number of words requested

  • Line 28: what exactly do you mean with “natural” variation? I think it is better to explicitly state what type of variation you would expect, for example frequency ranges, amplitude differences, song production rates and so on.

R: We call “natural variations” those that would occur in non-urbanized environments, we clarify this in introduction, because we use the concept from the title and along the manuscript. However, thanks to your observation we understand that the inclusion of the word in the hypotheses was not necessary.

  • Line 73: what do you mean “hardest urban core” here and in line 80? Perhaps change with “highly urbanised areas”

R: We are in agree, we have specified the concept to "heavily urbanized areas".

  • Line 83: would be better to state you time unit, for example “songs produced per hour” or “…per minute”

R: From now on, we have made the concept more precise using: "number of songs per individual per hour".

  • Line 84: the choruses do not “face” the soundscapes, they are rather “produced” or “occur” within certain urban soundscapes

R: We are in agree with the reviewer. We changed for “occur in”.

  • Line 85: still, not clear what you mean with “natural” variation. Of course there will be variation, but here you should clarify what type of variation you expect and why

R: We explained before the current use in the manuscript of the concept of “natural”, as well the changes introduced in the text.

  • Line 156-157: were the point counts conducted once per plot? Were they in the center of each plot? And at what time of the day were they conducted? Point counts should be conducted more than once, I would expect at least 3-4 at standard times or at different times during the day depending on the type of study

R: The counting points were at the center of the plot in all cases. Only Chiguanco Thrush individuals were counted for ten minutes, only in the morning period between 8:00 and 8:30. We evaluate each plot once and each morning we evaluate only one plot. It is true that subsampling each count point three or more times helps adjust the count point results, but our approach was to obtain as quickly as possible the approximate abundance of just this species by visiting each of the previously selected potential points (around of 20) in urban and extra-urban areas to select the final six study plots (three urban and three extra-urban). Our approach allows us to have this approximation of abundance in the same possible temporal conditions and with exactly the same effort. Individuals of this species are very conspicuous and active, so it is easy to count most individuals within 10 minutes. We are fairly confident that our abundance estimates allowed us to select plots with similar abundance. The data on the number of individuals registered on the recordings (Table 1, supplementary material) confirm this.

  • Line 194: what about territoriality in this species? During the breeding period males of many passerines, including trushes, sings within their territory, sometimes from selected perches. If that is so, you could have just controlled for this by taking GPS points of the recording and avoid recording multiple times in the same spot during subsequent visits

R: That is a very interesting suggestion. In each surveying day we evaded to recording the same individual as explained in methods. But when we visiting each plot the second occasion to record in the other chorus we obtained too a random sample of songs produced in the plot. The objective of the study was not to control the songs emitted by each individual, but rather to have a random sample of songs from several individuals from each population in each plot. For this same reason, it was not the objective to record the same individuals in the two periods of the day in each plot and compare the songs of each individual in the dawn and dusk choruses. This would serve for a study of characterizations of inter-individual and intra-individual variations within each population, and personality studies, which was not the objective of this study. Probably the confusion arises from a previous wording in the sentence in lines 233-235, as other reviewer point to us too, which seemed to say that we analyzed a certain number of songs from a certain number of individuals. In reality, we work only with the songs with the best signal and complete SSP and ESP data in the one-hour soundtrack, therefore, it is a random sample of songs from each population sampled for a certain period. We have modified this sentence to “After that, we selected for analysis only clear recordings of songs without other masking sounds and for which their respective SSP and ESP measurements were taken”.

  • Line 215-218: sure it is unlikely that the singing bird would maintain the same posture during the recording of SSP. However, here rather than “assuming” you could have done earlier preliminary tests. For example, you could use a recording (for example a song from Xenocanto) broadcasted from a speaker at a standard amplitude and change the position of the speaker in relation to the sound level recorder. This would tell you much variation in SSP values you should expect if a bird moves the head in the opposite direction of the recorder.

R: We are in agree with the reviewer, but head movements could vary more importantly if the microphone of the sound level indicator is unidirectional. For PCE-999 is omnidirectional (we are including this specification in the text as other reviewer suggest us). In this cases head movements produce less bias, but they exist anyway. Head movements occurs rapidly along the more or less three seconds each song is produced, and control them is very difficult. That is the reason we have performed the exact same method at all sites so that these biases do not align with any of our design factors.

  • Line 233-234: what you mean with “prone songs”? Before you said that it was impossible to know if you recorded the same individuals more than once, so how can you distinguish a song from a focal individual? I think here you mean only that you selected clear recordings of songs with no other background noise and with SSP and ESP measurements taken.

R: We agree, the sentence constructed before did not make clear what we wanted to say and only created confusion, we have changed it to “After that , we selected for analysis only clear recordings of songs without other masking sounds and for which their respective SSP and ESP measurements were taken”.

  • Table 2: please write the measurement units for the frequency values.

R: You are right, in table 2 we included Hz as units.

  • Line 489: change “diary” with “daily”

R: Done